# Increased Cytokine Levels in Seronegative Myositis: Potential Th17 Immune Response Implications

**DOI:** 10.3390/ijms252011061

**Published:** 2024-10-15

**Authors:** Andrea Aguilar-Vazquez, Efrain Chavarria-Avila, José Manuel Gutiérrez-Hernández, Guillermo Toriz-González, Mario Salazar-Paramo, Gabriel Medrano-Ramirez, Steven Vargas-Cañas, Oscar Pizano-Martinez, Cynthia-Alejandra Gomez-Rios, Christian Juarez-Gomez, José-David Medina-Preciado, Maribell Cabrera-López, Edgar-Federico Quirarte-Tovar, Ligia Magaña-García, Alejandra-Rubí García-Gallardo, Edy-David Rubio-Arellano, Monica Vazquez-Del Mercado

**Affiliations:** 1Centro Universitario de Ciencias de la Salud, Universidad de Guadalajara, Guadalajara 44340, Jalisco, Mexico; andrea.aguilarv@hotmail.com (A.A.-V.); christianjuarez147@gmail.com (C.J.-G.); 2Consejo Nacional de Humanidades, Ciencias y Tecnologías (CONAHCyT), Mexico City 03940, Mexico; 3Centro Universitario de Ciencias de la Salud, Instituto de Investigación en Reumatología y del SistemaMúsculo-Esquelético (IIRSME), Universidad de Guadalajara, Guadalajara 44340, Jalisco, Mexico; efrazan@gmail.com (E.C.-A.); oscar.pizano@academicos.udg.mx (O.P.-M.); cynthiagomez_9@hotmail.com (C.-A.G.-R.); 4División de Medicina Interna, Servicio de Reumatología, SNP-CONAHCyT, Hospital Civil Dr. Juan I. Menchaca, Guadalajara 03940, Jalisco, Mexico; maribellcabrera.17@gmail.com (M.C.-L.); edgarrquirartee@gmail.com (E.-F.Q.-T.); ligiamagana@hotmail.com (L.M.-G.); alejandra050799garcia@gmail.com (A.-R.G.-G.); 5Departamento de Disciplinas Filosófico, Metodológicas e Instrumentales, Centro Universitario de Ciencias de la Salud, Universidad de Guadalajara, Guadalajara 44340, Jalisco, Mexico; 6Laboratorio de Ciencias Básicas, Facultad de Odontología, Universidad Autónoma de San Luis Potosí, San Luis Potosí 78290, San Luis Potosí, Mexico; jose.manuel.gutierrez@uaslp.mx; 7Departamento de Madera, Celulosa y Papel, Centro Universitario de Ciencias Exactas e Ingenierías, Universidad de Guadalajara, Guadalajara 44340, Jalisco, Mexico; torizgmo@gmail.com; 8Departamento de Fisiología, Centro Universitario de Ciencias de la Salud, Universidad de Guadalajara, Guadalajara 44340, Jalisco, Mexico; msalazpa@hotmail.com (M.S.-P.); edavidrubio@gmail.com (E.-D.R.-A.); 9Departamento de Reumatología, Hospital General de México “Dr. Eduardo Liceaga”, Mexico City 06720, Mexico; gabrielmedranor@msn.com; 10Clínica de Nervio y Músculo, Departamento de Neurología, Instituto Nacional de Neurología y Neurocirugía “Dr. Manuel Velasco Suárez”, Mexico City 14269, Mexico; stevenvc@hotmail.com; 11Departamento de Morfología, Centro Universitario de Ciencias de la Salud, Universidad de Guadalajara, Guadalajara 44340, Jalisco, Mexico; 12Centro Universitario de Ciencias de la Salud, UDG-CA 703 Inmunología y Reumatología, Universidad de Guadalajara, Guadalajara 44340, Jalisco, Mexico; 13Unidad de Atención a Niñas, Niños y Adolescentes, Hospital Civil de Guadalajara Dr. Juan I. Menchaca, Guadalajara 44340, Jalisco, Mexico; drdavidmedina@hotmail.com; 14Departamento de Clínicas Quirúrgicas, Centro Universitario de Ciencias de la Salud, Universidad de Guadalajara, Guadalajara 44340, Jalisco, Mexico; 15Departamento de Ciencias de la Salud—Enfermedad como Proceso Individual, Centro Universitario de Tonalá, Universidad de Guadalajara, Guadalajara 45425, Jalisco, Mexico; 16Departamento de Biología Molecular y Genómica, Centro Universitario de Ciencias de la Salud, Universidad de Guadalajara, Guadalajara 44340, Jalisco, Mexico; 17Instituto Transdisciplinar de Investigaciones y Servicios (ITRANS), Universidad de Guadalajara, Guadalajara 45150, Jalisco, Mexico

**Keywords:** myositis, cytokines, IL-23, MYOACT

## Abstract

Th17 cells are known for producing IL-17 and their role in the pathogenesis of various autoimmune diseases, including myositis. Likewise, the participation of the IL-23/IL-17 pathway in autoimmunity has been confirmed. In this study, we aimed to evaluate the behavior of cytokines in myositis, focusing on the autoantibodies profile and the myositis core set measures. Twenty-five myositis patients were enrolled in this cross-sectional study. An expert rheumatologist evaluated the myositis core set measures. Serum levels of cytokines and chemokines were quantified using the LEGENDplex Multi-Analyte Flow Assay Kit from BioLegend. The autoantibodies detection was carried out using the line-blot assay kit Euroline: Autoimmune Inflammatory Myopathies from EUROIMMUN. We found higher serum levels of IL-33, CXCL8, IL-6, IL-23, and IL-12p70 in seronegative patients. A multiple linear regression analysis revealed that MYOACT scores could be predicted by the increment of IL-23 and the decrement of CCL2, IL-10, and CXCL8 serum levels. These findings suggest that the immune response in seronegative myositis patients exhibits an IL-23-driven Th17 immune response. The relevance of this discovery lies in its potential therapeutic implications. Insights into the IL-23-driven Th17 immune response in seronegative patients highlight the potential for targeted therapies aimed at modulating Th17 activity.

## 1. Introduction

Th17 cells are a subset of T-helper lymphocytes characterized by their participation in innate immunity in autoinflammatory and/or autoimmune rheumatic diseases such as seronegative spondylarthritis including psoriatic arthritis. IL-17 has been identified as the signature cytokine of Th17 cells. These cytokines recruit and activate other immune cells, amplifying the immune response [1].

Idiopathic Inflammatory Myopathies (IIMs), also known as immune-mediated myositis, are characterized by the presence of autoantibodies. However, seronegative myositis (negative for autoantibody detection by conventional methods) has been reported in up to 38.4% of patients in a group of almost 1700 patients from four different clinical cohorts. Even in the absence of a specific autoantibody, IIM manifests clinically cutaneous involvement, interstitial lung disease, and even pathognomonic clinical data such as heliotrope and Gottrön’s sign, which suggest the presence of an inflammatory process may be driven by cytokines, chemokines, and other molecules, even in the absence of these serological markers named autoantibodies [2]. In the context of seronegative myositis, it has been described as an increment of the Th17 and Th17.1 cell subpopulations [3].

The myositis autoantibodies are a unique field in autoimmune rheumatic diseases, since when these autoantibodies are classified as Myositis Specific-Autoantibodies (MSAs) that means they are exclusive to IIM patients, but also mutually exclusive. In other words, is infrequent that IIM patients express in their serum 2 or more MSAs (coexistence), which validates these autoantibodies as biomarkers of great value in the clinic [2]. Myositis Associated-Autoantibodies (MAAs) are different from MSAs in that they might also be present in other autoimmune rheumatic diseases. The MSAs are valuable tools for the rheumatologist for clinical classification, clinical prognosis, and even for making decisions therapeutic. Different clinical subgroups have been defined as part of the myositis spectrum, including dermatomyositis (DM), the most prevalent myositis in the world, amyopathic dermatomyositis (ADM), juvenile dermatomyositis (JDM), polymyositis (PM), inclusion body myositis (IBM), immune-mediated necrotizing myopathy (IMNM), juvenile myositis (JM) [4], cancer-associated myositis (CAM), and overlap myositis (OV) (coexistence of myositis and other connective tissue diseases) [5,6].

The corresponding myositis subgroup and clinical features for each MSA have been relatively well-described and different outcomes have been recognized; e.g., (i) anti-Mi-2 is associated with a classic DM, and usually with a good prognosis [7]; (ii) anti-MDA5 is commonly observed in rapidly progressive interstitial lung disease (RP-ILD) and/or vasculopathy in a clinical context of amyopathic or hypo-myopathic DM [8]; and (iii) anti-TIF-1γ and NXP-2 are considered as a high-risk factors for malignancy [9].

However, seronegative myositis represents a subset of patients who do not exhibit these typical autoantibodies. Seronegative myositis has been mainly described in the IMNM phenotype representing an IMNM profile with the worst clinical prognosis.

We must question ourselves which immunological mediators are implicated in the clinical manifestations in these seronegative myositis patients; we also question ourselves if there is a cytokine profile related to the absence of autoantibodies, the prevalent clinical manifestations, and their possible impacts on the clinimetry, that in the IIM field are referred as myositis core set measures.

According to the serological profile as seropositive or seronegative, the exploration of interleukins and chemokines involved in the Th17 pathway was carried out to address the possible influence in their clinical manifestations and status of activity or remission.

## 2. Results

### 2.1. Patients

We included 25 patients classified as having IIM according to Bohan and Peter [5,6], the 2017 European League Against Rheumatism/American College of Rheumatology (EULAR/ACR) classification criteria for adult and juvenile idiopathic inflammatory myopathies and their major subgroups [4], and European Neuromuscular Center (ENMC) criteria [10]. Seven men (28%) and eighteen women (72%) with a mean age of 44.1 ± 15.84 years with a mean disease duration of 4.6 ± 5.62 years, were included.

In our study, we included the subgroups of DM (n = 10), PM (n = 5), OV (n = 5), PM + anti-synthetase syndrome (n = 2), CAM (n = 2), and ADM (n = 1). Due to the small sample size of the subgroups, we only compared DM vs. PM as these two subgroups had a comparable number of seronegative patients.

### 2.2. Clinical Characteristics of Subgroups

Among the DM patients, 40% tested positive for anti-Mi2 and were clinically confirmed; DM is the most prevalent subgroup in our Mexican population as demonstrated by our group since 2013 [7]. Anti-TIF1γ and anti-SAE were present in 20 and 10 percent of patients; for both autoantibodies, the follow-up is specially oriented toward cancer screening [9]. The average disease duration of the DM patients was 5.8 years, with a range of 0 to 17 years. Treatment included methotrexate in 60% of patients, with doses ranging from 7.5 to 25 mg per week; prednisone in 90% of patients, with doses from 2.5 to 50 mg per day; hydroxychloroquine was used in 50% of patients at a dose of 200 mg per day; 10% were treated with mycophenolate mofetil at 1.5 g per day; and 30% of patients were treated with azathioprine at doses ranging from 50 to 100 mg per day. The average bilateral MMT8 was 139.1 (110 to 150), the average MYOACT score was 0.069 (0.00 to 0.370) and the average MDI score was 0.009 (0.00 to 0.027). An ADM patient was also included in the study with a disease duration of two years. They tested seropositive for anti-MDA5, were treated with methotrexate and hydroxychloroquine, their bilateral MMT8 was 150, their MYOACT score was 0.0, and their MDI score was 0.0. The mean of the CK (creatine kinase) level of the DM patients was 569.7 U/L with a range of 50 to 2888 U/L.

The PM patients recruited were all seronegative. Their average disease duration was 8.6 years, with a range of 0 to 18 years. Methotrexate was used in 60% of these cases at doses ranging from 7.5 to 25 mg per week and rituximab was used in 20% of these cases. The average bilateral MMT8 score was 135.8 (106 to 150), the average MYOACT score was 0.007 (0.00 to 0.033) and the average MDI score was 0.015 (0.00 to 0.036). The average CK level in PM patients was 150.75 U/L (35 to 376 U/L).

We included two patients with PM + anti-synthetase syndrome from recent onset. One of them was anti-EJ and the other was seropositive for PL-12. Both patients presented with pulmonary involvement. Their prescription was prednisone, mycophenolate, and/or methotrexate. The average bilateral MMT8 score was 95.5 (56 to 135), the average MYOACT score was 0.305 (0.283 to 0.327), and the average MDI score was 0.200 (0.118 to 0.283). The mean CK level was 2033 U/L (2024 and 2042 U/L).

Two patients in our study group presented with CAM. One of them was seronegative, developed a malignant tumor of the left apical lung, and was treated with methotrexate and prednisone. The other patient tested was a PM+ anti-synthetase syndrome positive for anti-EJ and anti-Ro-52. This patient developed a metastatic carcinoma with primary source being unknown and was treated with methotrexate and mycophenolate. The average bilateral MMT8 score was 149.3 (148 to 150), the average MYOACT score was 0.040 (0.000 to 0.117) and the average MDI score was 0.012 (0.000 to 0.027). The average CK level was 94.45 U/L (33 to 155.9 U/L). Unfortunately, both patients passed away.

Five patients were included in the OV group, where four patients presented sclerosis and myositis, named scleromyositis, and the other patient presented rheumatoid arthritis and myositis. Two were seronegative (40%), two were seropositive for anti-PMScl100 (40%), and one tested positive for anti-Ro-52 (20%). All of them were treated with methotrexate (100%), prednisone was used in 80% of patients, hydroxychloroquine in 60%, and mycophenolate in 20%. The average bilateral MMT8 score was 145.7 (120 to 150), the average MYOACT score was 0.026 (0.000 to 0.100), and the average MDI score was 0.015 (0.000 to 0.064). The mean CK level of this subgroup was 142.67 U/L with a range of 13 to 322 U/L.

### 2.3. Comparison between DM and PM Seronegative Patients

In total, we have 8/15 seronegative patients. Even with the small number of patients in this sub-analysis, we found higher MYOACT scores in seronegative DM patients (*p* = 0.036).

### 2.4. IL-10, IL-6, CXCL8, IL-1β, IL-33, IFN-γ, TNF-α, IL-23, and IL-17A Serum Levels Are Higher in Myositis than Healthy Subjects

We included 19 healthy subjects (HSs) as a reference group for the comparison of cytokine and chemokine serum levels. We found higher serum levels of IL-10, IL-6, CXCL8, IL-1β, IL-33, IFN-γ, TNF-α, IL-23, and IL-17A in myositis patients compared to the HSs (Figure 1).

From the HS data, we established a cut-off point for cytokine and chemokine serum levels. We considered a value as an outlier when the z score was higher than 1.96 (*p* = 0.005). We found several differences: fourteen patients (56%) presented IL-6 serum levels higher than the cut-off point, along with higher MYOACT scores (*p* = 0.012), higher LDH serum levels (*p* = 0.045), and an increased ESR (erythrocyte sedimentation rate) (*p* = 0.009). Thirteen patients (52%) showed increased CCL2 serum levels and lower MYOACT scores (*p* = 0.014). Eleven patients (44%) had IL-1β serum levels higher than the cut-off point with the same behavior as CCL2 patients, meaning high IL-1 β serum levels are also associated with lower MYOACT (*p* = 0.013) and MDI scores (*p* = 0.002). Nine patients (36%) had TNF-α serum levels higher than the cut-off point and higher aldolase levels (*p* = 0.039). Eight patients (32%) had IFN-α2 serum levels higher than the cut-off point with lower MYOACT (*p* = 0.045) and lower MDI (*p* = 0.027) scores. Eight patients (32%) had IFN-γ serum levels higher than the cut-off point and higher levels of aldolase (*p* = 0.033) and LDH (*p* = 0.035). Three patients (12%) had IL-23 serum levels higher than the cut-off point and lower MMT8 scores (*p* = 0.037). We found one patient (4%) with IL-18 serum levels higher than the cut-off point. We did not find differences in patients with outlier serum levels of CXCL8 (n = 14, 56%), IL-10 (n = 13, 52%), IL-33 (n = 10, 40%), IL-12p70 (n = 8, 32%), and IL-17A (n = 7, 28%) (Figure 1).

### 2.5. Cytokine Serum Levels Are Higher in Seronegative Myositis Patients: Potential Involvement of the Th17 Response

We compared the cytokine and chemokine serum levels between seropositive positive patients to any autoantibody vs. seronegative patients. We found higher serum levels of IL-33, CXCL8, IL-6, IL-23, and IL-12p70 in seronegative patients (Figure 2) with all of them belonging to Th17-immune microenvironment. It is interesting to highlight the same behavior of these biomarkers measured in the all the seropositive patients, being as it was almost undetectable in each of them.

Although we did not find higher serum levels of IL-17A in seronegative patients, we observed a positive correlation between the serum levels of IL-17A and the following cytokines: CXCL8 (rs = 0.673, *p* < 0.001), IL-33 (r_s_ = 0.635, *p* < 0.001), IL-12p70 (r_s_ = 0.549, *p* < 0.001), IL-6 (r_s_ = 0.530, *p* < 0.001), IL-23 (r_s_ = 0.460, *p* = 0.002), and IL-1β (r_s_ = 0.347, *p* = 0.021) (Appendix A). We did not observe any differences in clinical parameters or myositis core set measures between seronegative and seropositive patients.

Further analysis revealed that when we grouped our patients according to extra-muscular manifestations by organ and systems, higher IL-23 (*p* = 0.019) serum levels in DM patients with cutaneous involvement were observed (heliotrope and/or Gottron’s sign presence), as observed in 50% of seronegative patients.

### 2.6. The MYOACT Score Is Inversely Correlated with the Cytokines IL-1β and CCL2

We found a positive correlation between core set measures MYOACT and MDI (r_s_ = 0.484, *p* = 0.014), as well as a negative correlation between MYOACT and MMT8 (r_s_ = −0.513, *p* = 0.009), which denotes the increment of the disease activity.

We also found the following: (1) higher serum levels of IL-1β and CCL2 when the MYOACT score was equal to zero; (2) IL-1β and CCL2 are negatively correlated with the MYOACT score; (3) IL-1β serum levels are also negatively correlated with the MDI index (Figure 3).

## 3. Discussion

The relevance of seronegative status has been described in the IMNM subgroup [11]. In this phenotype, there are three well-defined profiles: anti-SRP positive, anti-HMGCR positive, and the seronegative subset. The worst prognosis, which is highly disabling, has been associated with seronegative IMNM patients [12,13,14]. In contrast, in seronegative ADM for anti-MDA5 patients, case reports described a better prognosis as they did not present lung involvement or vasculopathy and had a good response to immunosuppressive treatment [15,16]. These studies denote the heterogeneity and complexity of the outcomes of MSAs [17]. In seronegative patients, the detection of elevated cytokine serum levels could be an indicator of the presence of active inflammation despite the MSA/MAA absence [18,19].

In this study, we described the behavior of cytokines in myositis patients, focusing on the seronegative profile of autoantibodies and the myositis core set measures. We first described the behavior of IL-33, CXCL8, IL-6, IL-23, and IL-12p70 at higher levels in seronegative IIM patients (Figure 2).

According to our results, the elevated cytokines found in the myositis seronegative patients (IL-6, IL-23, IL-12p70, IL-33, and CXCL8) are related to a Th17 profile. IL-6 and TGF-β are the classical activating cytokines for a T-naïve cell for Th17 cell differentiation. Meanwhile, IL-23 is also critical for Th17 maintenance [20]. IL-12 has been described as the optimum cytokine to expand human Th17 cells in vitro [21]. IL-33 is commonly associated with Th2 cell induction; however, it can also stimulate Th17 cells in an autoimmune microenvironment by stimulating IL-1 and IL-6 [22]. Finally, CXCL8, a chemokine responsible for neutrophil recruitment, is also associated with Th17 immune responses; Th17 cells produce IL-17A, which can induce the production of CXCL8, and both cytokines have been observed as elevated in asthma and other inflammatory diseases, suggesting they could promote the sustained pro-inflammatory microenvironment [23]. Therefore, we suggest a possible differential immune response.

On the other hand, we observed a positive correlation between the serum levels of IL-17A and the following cytokines: CXCL8, IL-33, IL-12p70, IL-6, IL-23, and IL-1β (Figure 3), which remarks the possibility of our hypothesis that behind seronegative myositis there is an immune response orchestrated by Th17-immunity.

Th17 cells represent a subset of pro-inflammatory T-helper cells. It has been reported that there is a higher percentage of Th17 and Th17.1 subsets in seronegative myositis [3].

We observed higher serum levels of IL-23 when heliotrope and/or Gottron’s sign was present (*p* = 0.019), as observed in 50% of seronegative patients. As we know, IL-23 is a heterodimeric cytokine composed of the IL-12p40 subunit and p19 subunit, is part of the IL-12 family, and is a key participant in the expansion of Th17 cells. It is primarily secreted by dendritic cells (DCs) and activated macrophages found in the peripheral organs (skin, intestinal mucosa, and lung). These cells are highly pro-inflammatory and play a crucial role in the pathogenesis of seronegative spondylarthritis [24,25]. A psoriatic arthritis (PsA) mouse model was induced by overexpressing IL-23 in the skin (K23 mice) [26]. Therefore, IL-23 levels in this context, seem to be related to the cutaneous findings. In a murine autoimmune myositis model, the severity of myositis was rescued by anti-IL-23R antibodies [27]. Nowadays, IL-23 has been recognized as a therapeutic target for inflammatory myopathy [27] [28]. However, the importance of the IL-23/IL-17 in myositis has not been thoroughly characterized and there are still unexplored areas as the probable relation between Th17 cells and seronegative myositis.

When we analyzed the cytokines behavior according to the myositis core set measures, we observed an association with CCL2 and IL-1β, showing higher serum levels when the MYOACT score is equal to zero, which might be interpreted as clinical remission. In this regard, it has been recently reported that there is a significant difference in CXCL9, lymphotoxin-α, MMP-3, MCP-1, and IL-8, between patients with active versus inactive disease [29,30].

CCL2 elevation has also been associated with the muscle regeneration process [31,32,33], stimulation of monocytes to produce pro-inflammatory cytokines, the enhancement of macrophages, and neutrophils’ survival, as well as the macrophage polarization to M1 or M2 [34].

CCL2 has a prominent role in the muscle microenvironment since it seems to trigger muscle fiber regeneration [31,35,36]. It has been demonstrated that human myogenic precursor cells (MPCs), also known as satellite cells, can secrete CCL2. This chemotactic activity declines according to the MPC differentiation [36]. Another finding about CCL2 and muscle regeneration is a murine model C57Bl/6J where the CCL2 gene was silenced; in this study, after muscle injury caused by ischemia, impaired muscle regeneration was reported [31]. CCL2 has been found colocalized in muscle regeneration with CD56 (a marker of myogenic cells, satellite cells, myoblast, and regenerating muscle fibers) [36].

It has been reported that there is increased CCL2 in serum and muscle tissue [37,38]. Some studies have related CCL2 with the pathophysiology of ADM and ILD [39,40]. In JDM, a correlation with disease duration has been described [41]. Considering all our findings, we suggest the immune response involved in seropositive myositis patients belongs to the Th2 subset, meanwhile the immune response in seronegative myositis patients is orchestrated by the Th17 subset.

When the treatment prescribed to these patients, independent of the serological status, was analyzed, we found that when glucocorticoids (GCs) were not prescribed, IL-33 and IL-17A were increased as Th17 pathway-involved molecules, which might be interpreted as the absence of non-genomic effects of GCs and more proinflammatory responses in these cases. CCL2 was found to be higher in patients not taking HCQ; since its mechanism of action is to avoid the Toll-like Receptor (TLR) expression, in particular TLR9 and TLR7, it might be the inflammatory pathway that these patients might have not interfered. Finally, even some authors reported that azathioprine does not modify the levels of IL-6 in rheumatoid arthritis patients; we found that IL-6 levels were higher when azathioprine was not prescribed [42] (Appendix A).

The exploration of interleukins and chemokines in this research work suggests a Th17 cytokine profile in seronegative myositis patients, which might be implicated in the pathogenesis of this subset. According to our results, the elevated cytokines found in the myositis seronegative patients (IL-6, IL-23, IL-12p70, IL-33, and CXCL8) are related to a Th17 profile.

IL-6 and TGF-β are the classical activating cytokines for a T-naïve cell differentiation into Th17 subpopulation. IL-6 and TGF-β promote the transcription factor RORγt expression, which in turn enhances the IL-17 and IL-23 receptor expression. Therefore, IL-23 plays a critical role in the maintenance, expansion, and survival of Th17 cells through a positive-feedback loop that enhances the expression of IL-17, RORγt, and other pro-inflammatory cytokines, including TNF, IL-1β, and IL-6, ensuring sustained Th17 cell function and population stability [20,43].

In addition, IL-12 shares its p40 subunit with IL-23 and has been described as another cytokine able to expand human Th17 cells in vitro [21]. IL-33 can also act as a positive feedback for Th17 cells in an autoimmune microenvironment by stimulating IL-1 and IL-6 [22]. Finally, IL-17A can induce the production of CXCL8, suggesting they could promote the sustained pro-inflammatory microenvironment [23].

Therefore, we suggest a possible differential immune response in seronegative myositis mediated by an induction of the Th17 inflammatory response, through IL-6, with maintenance of the Th17 pathway by IL-23 and IL-33. Meanwhile, an interesting finding in patients in clinical remission (MYOACT = 0) suggests that CCL2 and IL-1β might be inversely correlated as the muscle regeneration process is ongoing.

This work encourages the scientific and clinical community to continue investigating the pleiotropic and heterogeneous implications of immune mediators in the field of myositis. The relevance of these findings lies in future treatment targets. The autoantibodies’ presence commonly represents a guide for clinical phenotype, muscular and extra-muscular involvement, and prognosis; however, its absence (seronegative) could make the medical approach unclear. For this reason, we believe it is essential to understand seronegative myositis to develop tailored therapy and enhance patient outcomes.

### Limitations

The major caveat of this study was the small sample size with many types of myositis, such as OV, CAM, and ASS; however, the findings were significant (*p* < 0.005) and the size effect measured reached medium to high values.

## 4. Materials and Methods

### 4.1. Patients

Twenty-five patients classified with myositis according to Bohan and Peter [5,6], the 2017 EULAR/ACR criteria [4], and ENMC [10] criteria were enrolled in this cross-sectional study. We also included nineteen HSs for cytokine and chemokine serum levels comparison; they were paired by age and sex and were recruited from the same Mexican population as the myositis patients.

Informed written consent was obtained from every subject before enrollment in the study. This protocol was approved by the Ethical and Research committees of Hospital Civil de Guadalajara “Dr. Juan I. Menchaca” (Secretaría de Salud Jalisco registration number 0318/19 HCJIM/2019).

We obtained peripheral blood from all patients. Serum was obtained by centrifugation of the blood samples (3500 rpm, 10 min) and stored at −20 °C until analysis. An expert rheumatologist evaluated the myositis core set measures; likewise, all methods were performed following the relevant guidelines according to a validated protocol.

### 4.2. Activity Assessment

Disease activity is defined as potentially reversible clinically evident pathology or physiology resulting from the myositis disease process. It was evaluated using the validated MYOACT score from the Disease Activity Core Set Measure proposed by the International Myositis Assessment and Clinical Studies Group (IMACS) [44].

### 4.3. Damage Assessment

According to IMACS, damage is defined as persistent changes in anatomy, physiology, pathology, or function, which are present for at least 6 months. Disease damage was assessed by employing the MDI from the Disease Damage Core Set Measure [44].

### 4.4. Muscle Strength Assessment

It was performed using the MMT8 score. This tool includes seven proximal muscle groups which are tested bilaterally (deltoids, biceps, wrist extensors, quadriceps, ankle dorsiflexors, gluteus medius, and gluteus maximus) and neck flexors, which complete the eight muscle groups tested. Each muscle subgroup receives a score of ten points obtaining a maximum score of 150 when the patient did not present muscle weakness [45].

### 4.5. Autoantibodies Detection

It was carried out in the serum samples of the IIM patients using the line-blot assay kit Euroline: Autoimmune Inflammatory Myopathies from EUROIMMUN Medizinische Labordiagnostika AG (31 Seekamp. Lübeck, DE 23,560, Germany), according to the test instructions. This kit allows the detection of twelve MSAs (anti-Mi-2α, Mi-2β, TIF1γ, MDA5, NXP2, SAE1, Jo-1, SRP, PL-7, PL-12, EJ, and OJ) and four MAAs (Ku, PM-Scl100, PM-Scl75, and Ro-52). This kit includes test strips coated with highly purified and biochemically characterized autoantigens which were incubated with diluted serum samples, and in the case of positivity, the autoantibodies were bounded to the corresponding site and catalyzed a color reaction. For the correct evaluation of the test strips, they were analyzed by the EUROLineScan software version 3.1, and the results were interpreted according to the signal intensity of the bands.

### 4.6. Cytokine and Chemokine Serum Quantification

Serum levels of the cytokines IL-1β, IFN-α2, IFN-γ, TNF-α, IL-6, IL-10, IL-12p70, IL-17A, IL-18, IL-23, and IL-33 and chemokines CCL2 and CXCL8 were quantified using the LEGENDplex Multi-Analyte Flow Assay Kit from BioLegend (9727 Pacific Heights Blvd. San Diego, CA, USA) following test instructions. This multiple bead-based immunoassay is based on the basic ELISA-type sandwich-assay principle. Each bead is coated with a specific antibody for the cytokine or chemokine of interest. Subsequently, these beads were differentiated by size and internal fluorescence intensity using a flow cytometer (Applied Biosystems, Attune NxT). Measures were analyzed using the online software LEGENDplex https://legendplex.qognit.com/ (accessed on 1 November 2023).

### 4.7. Statistics

Values are presented as mean ± standard deviation (SD), median with interquartile range (IQR), or percentages (%), as appropriate. Comparisons were made using the Mann–Whitney U test with Fisher’s exact test for quantitative variables, due to the low number of patients. Meanwhile, qualitative variables were analyzed using χ2 or Fisher’s exact test as appropriate. A multiple linear regression analysis was carried out with backward mode and *p* = 0.05 for entrance and *p* = 0.10 for elimination.

Our data were analyzed using SPSS version 24.0 software (SPSS Inc. Chicago, IL, USA) and GraphPad Prism version 6.00 for Windows (GraphPad Software, La Jolla, CA, USA), considering a two-tailed level of *p* < 0.05 to be significant for analysis. The size effect was calculated by the rank–biserial correlation, considering a small effect when r ≈ 0.1, medium effect when r ≈ 0.3, and high effect when r > 0.5.

## Figures and Tables

**Figure 1 ijms-25-11061-f001:**
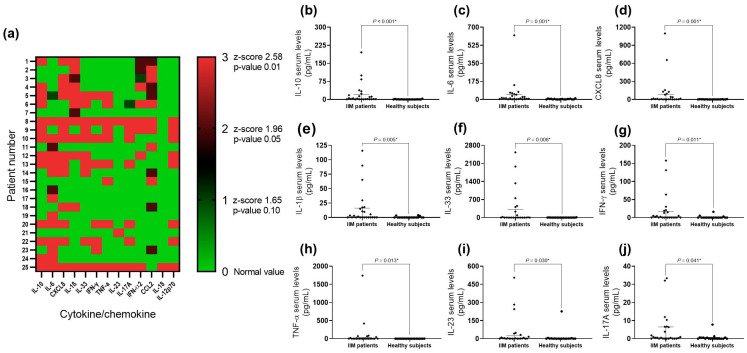
Cytokine and chemokine levels in myositis patients contrasted to healthy subjects; higher serum levels of IL-10, IL-6, CXCL8, IL-1β, IL-33, IFN-γ, TNF-α, IL-23, and IL-17A in myositis patients were observed. (**a**) Heatmap of cytokine serum levels according to z-score in each patient; (**b**) IL-10 (*p* < 0.001, r = 0.7); (**c**) IL-6 (*p* < 0.001, r = 0.5); (**d**) CXCL8 (*p* < 0.001, r = 0.5); (**e**) IL-1β (*p* < 0.001, r = 0.4); (**f**) IL-33 (*p* < 0.001, r = 0.4); (**g**) IFN-γ (*p* < 0.001, r = 0.4); (**h**) TNF-α (*p* = 0.013, r = 0.4); (**i**) IL-23 (*p* = 0.030, r = 0.3); and (**j**) IL-17A (*p* = 0.041, r = 0.3). * Mann–Whitney U test with Fisher’s Exact test. The size effect was calculated by the rank–biserial correlation.

**Figure 2 ijms-25-11061-f002:**
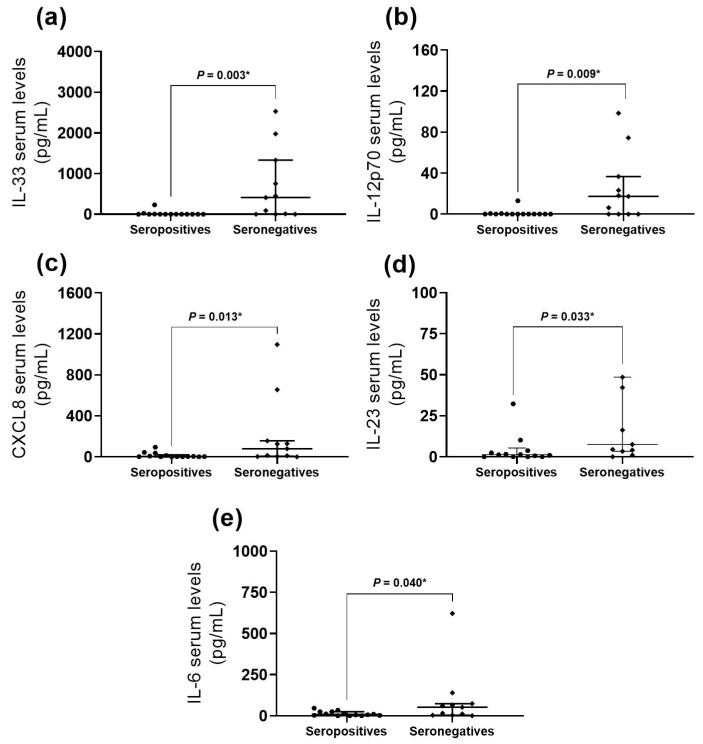
Cytokine and chemokine levels according to autoantibodies presence. Higher serum levels of the following autoantibodies were detected in the seronegative myositis subset: (**a**) IL-33 (*p* = 0.003, r = 0.6); (**b**) IL-12p70 (*p* = 0.009, r = 0.5); (**c**) CXCL8 (*p* = 0.013, r = 0.5); (**d**) IL-23, (*p* = 0.033, r = 0.4); and (**e**) IL-6 (*p* = 0.040, r = 0.4). * Mann–Whitney U test with Fisher’s Exact test. The size effect was calculated by the rank–biserial correlation.

**Figure 3 ijms-25-11061-f003:**
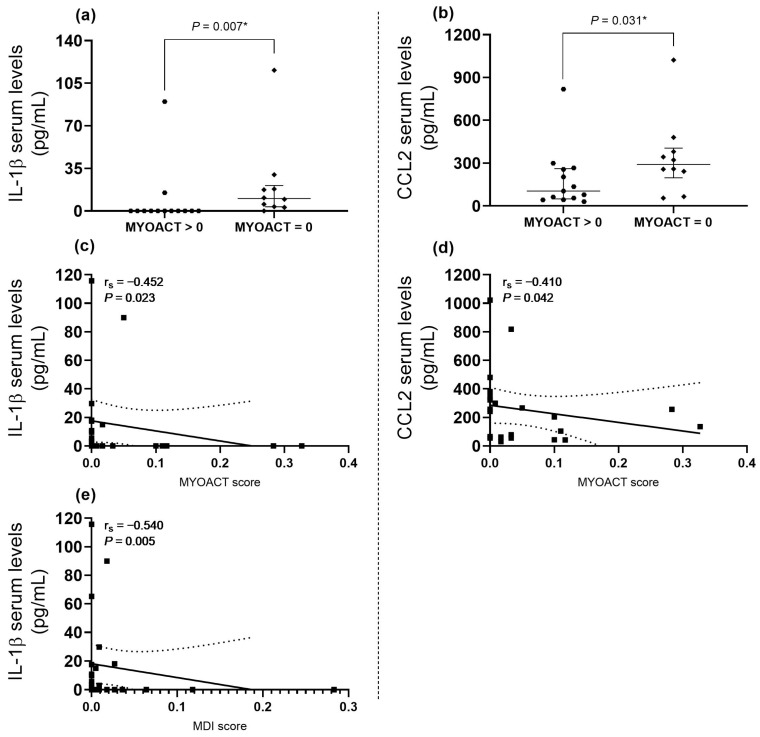
Cytokine and chemokine serum levels according to core set measures in myositis patients: (**a**) higher serum levels of IL-1β in MYOACT > 0 (*p* = 0.007, r = 0.5); (**b**) higher serum levels of CCL2 in MYOACT > 0 (*p* = 0.031, r = 0.4); (**c**) negative correlation between IL-1β serum levels and MYOACT score (*p* = 0.023, r_s_ = −0.452); (**d**) negative correlation between CCL2 serum levels and MYOACT score (*p* = 0.042, r_s_ = −0.410); and (**e**) negative correlation between IL-1β serum levels and MDI score (*p* = 0.005, r_s_ = −0.540). * Mann–Whitney U test with Fisher’s Exact test. The size effect was calculated by the rank–biserial correlation.

## Data Availability

The original data presented in the study are openly available in FigShare at https://figshare.com/s/546463c508d2067c85b6 (accessed on 11 September 2024).

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
