# Peer review of "Increased Cytokine Levels in Seronegative Myositis: Potential Th17 Immune Response Implications"

_ijms, 2024, doi:10.3390/ijms252011061_

Round 1
Reviewer 1 Report
Comments and Suggestions for Authors
I congratulate the authors, the work is very interesting, well structured and well written and scientifically valid. I have only a few minor remarks.
Minor questions:
1) In the results section (lines 121 - 159) the authors describe the characteristics of the cohort of patients studied, including the therapy they were undergoing and the clinical scores (MYOCAT, MDI and MMT8). In my opinion the authors should also add data relating to serum creatine kinase (CK) levels, since this is one of the main blood-chemical indices of muscle damage. If the data, such as the CK level at diagnosis and/or before the start of drug treatment is known, this data should be added in the results section. Also to evaluate any differences between cases of seronegative vs seropositive myositis also in relation to cytokine levels.
2) In the results section the authors describe the cytokine levels analyzed in the patients affected by myositis considered in the present study. It would be very interesting if data on the level of Th17 analyzed on peripheral blood samples (of the studied patients) had also been collected. Data of this type, taken at the time of diagnosis or before the beginning of therapy, or in any case at the time of blood sampling for the analysis of serum cytokine levels would be a very important data. If a cytometric quantification of Th17 (for example evaluating the percentage of CD4 memory IL-17A positive) made on peripheral blood or using other markers associated with Th17 such as the quantification of CD4/CD45RO/CCR6 positive CXCR3 negative cells has been done, these data should be added to strengthen the work and make it stronger and more interesting.
3) On page 5 (figure 1) the authors show the comparison of cytokine levels between patients and healthy controls. I would suggest the authors to slightly increase the name on the Y-axis of each graph in figure 1, to make the writing more readable since it appears small.
4) As recommended for figure 1, I would also suggest for figure 2 to slightly increase the size of the caption on the Y-axis, of each graph in figure 2.
5) As for figure 1 and 2 also for figure 3 I would increase the size of the legend on the Y-axis of the graphs.
6) In the discussion section (lines 349 - 354) the authors describe the importance of IL-17 in myositis although the importance of the IL-23/IL-17 axis is not fully understood in this type of pathology. I would suggest the authors to add two lines of comment on the combined effect of IL/17 and IL-23 and their partway. That is, how IL-23 promotes the differentiation of Th17 (after interaction with its receptor, expressed on these lymphocytes) promoting their activation and the production of IL-17 by them. All via activation of STAT3 which is central as a transcription factor for both IL-23 and IL-17 activity. Two lines of comment are enough.
Author Response
Please see the attachment.
All changes to the manuscript have been highlighted in yellow.

Reviewer 2 Report
Comments and Suggestions for Authors
The last 2 paragraphs from the Introduction section belong to the Results section (they contain results).
Results, 2.4:
VSG abbreviation is not explained
TFN-alpha is probably TNF-alpha (line 177)
Given the study's objectives, this section makes no sense (comparing patients with healthy subjects). Moreover, we suppose some cut-off values for all these cytokines have already been established in other studies on more than 19 healthy subjects.
We were not convinced by the rationale behind comparing seropositive with seronegative in a heterogenous group of diseases (although all are inflammatory myopathies, we know their pathogenesis is different). With such a small sample, there are even smaller samples for each myopathy form, further diminishing the validity of the analysis.
However, the differences look impressive (Fig 2), at least graphically, as no effect measure is given (only p-values). Therefore, this exploratory study probably needs to be continued to validate some hypotheses (if true).
This study contains multiple comparisons and correlations of many variables; therefore, many of the significances could appear by hazard. Moreover, it is hard to depict the importance of the found correlations between the inflammatory cytokines.
Also, the activity scores for myopathies are expected to be correlated (the lack of correlation would have signaled a problem with these scores).
“To further assess the effect of the cytokines on MYOACT and MDI, we performed a multiple linear regression analysis with the serum levels of cytokines log-transformed for data linearization adjusted by disease duration. For this analysis, we included the seropositive or seronegative status, IL-33, CXCL8, IL-6, IL-23, and IL-12p70 (cytokines that were elevated in seronegative patients) and IL-10 as a reference.”
1. We are not sure that the model is correct, as the dependent variable is not quantitative but ordinal (a composed score).
2. Additionally, the paper doesn't provide evidence that the assumptions of multiple linear regression (such as linearity, homoscedasticity, normality of residuals, and independence of errors) were checked. Given the small sample size, the regression results could be particularly vulnerable to violation of these assumptions. This issue should be explicitly addressed in the paper.
3. The authors should consider the meaning of the fact that seropositivity was not retained in the model but only the cytokine levels. That contradicts the hypothesis that seropositivity leads to a different phenotype/prognostic of the disease and the search for pathogenetic differences (like cytokine level) that was the initial objective.
Also, the authors should Include a more detailed analysis or discussion of how immunosuppressive treatments (e.g., corticosteroids, methotrexate) may affect cytokine levels, and how these could have influenced the study’s findings.
In the end, I think the objectives of this research should be more clearly stated.
Author Response

(The authors gave the same response as above.)

Round 2
Reviewer 2 Report
Comments and Suggestions for Authors
DIscussion section is very long, and it lacks a limitations section (small sample size with many kinds of myositis, multiple comparisons). No effect sizes are given, only p-values.
Author Response

(The authors gave the same response as above.)

Round 3
Reviewer 2 Report
Comments and Suggestions for Authors
The authors made the requested changes